# Nuclear energy acceptance in Poland: From societal attitudes to effective policy strategies —Network modeling approach

**Pawel Robert Smolinski**[1]*, **Joseph Januszewicz**[2], **Barbara Pawlowska**[1],
**Jacek Winiarski**[1]

**1** University of Gdansk, Gdansk, Poland, **2** Dartmouth College, Hanover, NH, United States of America

* pawel.smolinski@phdstud.ug.edu.pl

## Abstract

Poland is currently undergoing substantial transformation in its energy sector, and gaining public support is pivotal for the success of its energy policies. We conducted a study with 338 Polish participants to investigate societal attitudes towards various energy sources, including nuclear energy and renewables. Applying a novel network approach, we identified a multitude of factors influencing energy acceptance. Political ideology is the central factor in shaping public acceptance, however we also found that environmental attitudes, risk perception, safety concerns, and economic variables play substantial roles. Considering the long-term commitment associated with nuclear energy and its role in Poland's energy transformation, our findings provide a foundation for improving energy policy in Poland. Our research underscores the importance of policies that resonate with the diverse values, beliefs, and preferences of the population. While the risk-risk trade-off and technology-focused strategies are effective to a degree, we advocate for a more comprehensive approach. The framing strategy, which tailors messages to distinct societal values, shows particular promise.

## Introduction

Renewable energy has seen a significant rise in acceptance around the globe as an effective method of combating climate change. Many countries, along with international organizations like the United Nations and European Union, have set ambitious agendas for carbon neutrality. These plans primarily rely on the extensive deployment of renewable energy resources. However, the landscape of energy transition continues to be a subject of debate among scholars and policymakers. While some assert that a complete energy transition could be achieved through renewables without the aid of nuclear energy [1, 2], others dispute this perspective [3, 4]. Instead, they posit that a holistic energy transition necessitates the inclusion of more stable and predictable power sources, such as nuclear energy, in tandem with renewables [4–6].

Nuclear energy, recognized for its longevity, often represents a significant long-term financial commitment given the typical lifespan of a nuclear power plant, which could exceed 80

**Data Availability Statement:** The supplementary materials are accessible via the following link: https://osf.io/neagr/. The supplementary materials

include: • Dataset: This contains the raw data used in the analysis. • Supplementary explanations: This document provides a breakdown of the code and additional explanations related to the model used in the analysis. • Supplementary tables: This document contains supplementary tables with additional data and information related to Figs 2 and 3 in the main text. • R script: This is the R script containing the code used for data analysis and generating the results and figures.

**Funding:** This research was funded by the University of Gdansk, Faculty of Economics. Internal financing.

**Competing interests:** The authors declare that they have no known competing financial interests or personal relationships that could have appeared to influence the work reported in this paper.

years. Consequently, investigating and securing public acceptance of nuclear energy is a pivotal part of any strategic investment plan in this sector [7]. When public acceptance is misaligned with energy policies, countries can face setbacks, as seen with Germany's premature decommissioning of its nuclear facilities in 2023 [8].

This paper investigates the determinants of nuclear energy acceptance in Poland, a nation currently experiencing a shift towards nuclear power. Through a detailed analysis of public attitudes, we pinpoint the core determinants that policymakers and stakeholders should consider when formulating effective nuclear energy policies. Importantly, while our focus is on Poland, the insights from our research have broader implications. They offer valuable direction for international bodies and governments worldwide, especially in contexts where public acceptance of nuclear energy remains unclear.

## Literature review

### Public acceptance

Public acceptance of energy technologies, as defined by Upham et al. [9], pertains to "the attitude or behavioral response to the implementation or adoption of a proposed technology held by the lay public of a given country, region or town". The evaluation of public behavioral response is difficult due to the lack of direct usage or investment of nuclear energy by the average person. However, attitudes can be assessed by measuring public opinion on nuclear energy viability as an investment (investment attitude) and gauging the perceived trade-off between associated risks and benefits (risk-benefit trade-off).

To understand public attitudes, we consider factors such as political ideology and safety concerns as well as environmental and economic variables. This segment consolidates findings from global acceptance research, with an emphasis on determinants influencing the acceptance of nuclear energy.

### Political ideology

Previous studies on nuclear energy indicate that political ideology significantly influences its acceptance. Surveys conducted in the United States revealed that people's political attitudes dictated their acceptance of different energy sources. Specifically, conservatives tended to favor nuclear energy, while liberals were more inclined towards renewable sources [10–12].

In a more recent study in South Korea, Chung and Kim (2018) found that acceptance of nuclear power was often associated with conservative political values, suggesting that energy policy is as much a political issue as it is a scientific or economic one. This political dimension is a common thread across countries, underscoring the need for a politically sensitive approach to nuclear energy policy. Political ideology is also a significant influencing factor for renewable energy acceptance, as evidenced by studies from Europe [13], the United States [10, 14] and Canada [15]. This central role of political attitudes in determining energy acceptance, irrespective of the energy source, suggests a potential polarization issue. This divide may manifest as conservatives leaning towards nuclear energy as a solution to climate change, while liberals might favor renewables [16]. Nevertheless, the issue of polarization in energy acceptance still remains under-explored in academic research.

### Environmental and climate attitudes

The public acceptance of nuclear energy is likely influenced by environmental and climate attitudes. Recently, Hu et al. [17] found that environmental concern and energy shortage belief significantly impacted public acceptance of nuclear energy in China. Similarly, a previous

study involving Chinese college students demonstrated a positive relationship between environmental awareness, environmental safety, and nuclear power acceptance [18]. Conversely, studies from Europe suggest that individuals with pro-climate attitudes tend to oppose nuclear power development [19, 20]. There is a need to address these discrepancies by examining the ties between political views, environmental attitudes, and the acceptance of nuclear energy. This need is highlighted in the earlier section discussing the relationship between political ideology and energy acceptance, and the recognized correlation between liberal views and environmental concerns [21].

## Risk perception and safety concerns

Risk perception and safety concerns have been established as major influencing factors in the acceptance of nuclear energy across nations [22]. However, the risk concerns might not be directly tied to power plant operation safety or fears of nuclear accidents. Instead, Sjoberg and Drottz-Sjoberg [23] suggest that the public is more concerned about waste management and the environmental impact of nuclear energy. This perspective was recently supported by Temper et al. [24], who found that many anti-nuclear protesters worldwide primarily highlight issues related to waste management and environmental pollution. Similarly, Pidgeon et al. [25] suggest that the emphasis on power plant operation safety might not be as influential in determining public acceptance as previously believed, because the public's concerns about nuclear energy are more focused on the long-term environmental impacts, rather than operation risks and safety issues. Therefore, a comprehensive understanding of nuclear energy acceptance requires examining risk tradeoffs, operation safety, environmental concerns, and waste management attitudes collectively.

## Economic considerations

Besides environmental and safety attitudes, research highlights the significance of economic factors, such as cost-effectiveness and efficiency [26, 27]. A report by Schrems et al. [28] illustrated that public perceptions about nuclear energy prices, compared to renewable alternatives, significantly impact its acceptance. The argument presented in the report, that nuclear energy prices were significantly higher than those for renewables, was influential in public campaigns promoting Germany's Energiewende policy. Given the critical role of economic arguments in shaping public policy, we must consider their effects on public acceptance, especially in nations like Poland, currently transitioning towards nuclear energy.

Further, the recent geopolitical landscape, exemplified by the 2022 Russian invasion of Ukraine and subsequent EU sanctions, has amplified the European emphasis on energy independence. Significant research efforts are underway to optimize the design of next-generation nuclear reactors, enhancing safety, efficiency, and sustainability [29–31]. This focus, whether on self-reliance, on partnering with reliable allies or on next-generation reactors has made it crucial to measure public attitudes on nuclear energy as means to bolster national energy independence.

Synthesizing literature on public acceptance of energy technologies reveals numerous influencing factors, including political, environmental, and economic. While individual analysis of these factors is valuable, a holistic understanding of their collective influence is necessary for informed policy implementation.

## Public acceptance and strategies for policy implementation

The effectiveness of energy transition policies depends largely on public acceptance. Policies that align with the values, beliefs, and preferences of the majority are more likely to be viewed

as legitimate and fair [32]. When the public perceives policies as legitimate, they are more likely to comply with them voluntarily, reducing the need for enforcement actions and potential conflicts. Public acceptance ensures smoother execution because citizens are more likely to cooperate and participate actively. Conversely, policies that face public resistance can encounter delays, increased costs, and potential failures in achieving their intended outcomes [33]. This stability is crucial for long-term planning, especially in sectors like energy, environment, or infrastructure, where projects span decades.

To circumvent such outcomes, governments must formulate strategies that foster favorable attitudes toward the important energy technologies. Social change initiatives are instrumental in this regard, wherein governmental bodies advocate for the policy, aiming to elevate its widespread acceptance. Contemporary strategies include the trade-off approach, which emphasizes the positive trade-off between benefits accompanying nuclear energy and its associated risks [25], and informative strategies, which rely on disseminating information about specific facets of energy technology [17]. Emerging strategies, such as framing and nudging, also hold promise in improving acceptance [34, 35]. We posit that an in-depth analysis of the national acceptance of nuclear energy can offer insights into the strengths and limitations of available strategies, thereby steering policymakers toward optimal decision-making.

With the understanding that policies aligned with public values have greater success and longevity, it becomes imperative to contextualize this in specific regions currently undergoing energy transitions. One such nation is Poland, thereby offering a case for understanding how nuclear energy acceptance informs evaluation of policy strategies. The next section provides insights into Poland's energy transition and the context for nuclear energy acceptance therein.

## Background: Poland's energy transition and nuclear energy acceptance

Poland is actively transitioning its energy sector in line with the Paris Agreement and the European Green Deal, targeting net-zero greenhouse gas emissions by 2050. The nation commits to raising renewable energy sources (RES) to 21–23% of its final energy consumption by 2030 and decreasing coal's role to 56–60% in electricity production [36]. The "Polityka energetyczna Polski do 2040 roku" policy highlights Poland's dedication to energy independence, efficiency, and safety. As the country shifts from coal to renewables and nuclear energy, challenges such as substantial investment financing and ensuring the highest safety standards for nuclear plants persist.

However, beyond these technical and financial challenges lies the crucial aspect of public acceptance. As highlighted in the literature review, public acceptance of nuclear energy is a complex issue influenced by various factors. In Poland, the historical, environmental, and political contexts play significant roles in shaping public opinion.

Recent surveys conducted by the Polish Centre for Public Opinion Research (CBOS, Polish: Centrum Badania Opinii Społecznej) between 2009 and 2018 highlight an increasing awareness of climate change among the Polish population. This awareness is accompanied by an expectation for a decline in the reliance on coal and an increased endorsement of renewable energy sources. However, there remains a significant reluctance in Poland to adopt nuclear energy. This caution can be traced back to the 1970s. During this period, the government of the Polish People's Republic initiated plans to build a nuclear facility in Żarnowiec using Soviet technology. Following the tragic events of Chernobyl in 1986, Poland experienced strong protests that ultimately led to the discontinuation of the Żarnowiec project by 1990 (Jaszczak, 2022).

Over the past 15 years, CBOS research indicates that the Polish sentiment towards the construction of nuclear power plants was largely negative. Opposition peaked in 2006 with 58% opposing their construction. By 2009, the attitudes seemed to shift slightly in favor of nuclear energy, however his support was short-lived as the 2011 Fukushima disaster brought about a decade-long reversal in public opinion in Poland. It was the new government policy and the 2022 energy crisis in Europe that invigorated debate on nuclear energy.

According to the results of a survey conducted in November 2022 by CBOS on the attitude of Poles to nuclear energy, 54% of respondents declare support for the construction of nuclear power plants in Poland [37]. A positive attitude to the development of nuclear power dominates both among respondents with right-wing (83%) and centrist (76%) political views. There was a reduction in differences in support by age group. Whereas earlier rather young people opted for this form of energy, now there are no such differences. Poles also believe that investing in the development of nuclear energy is necessary if they want to move away from coal-based energy—this opinion is shared by more than three quarters of the respondents (76%). In May 2021, such responses were recorded at 44%. There is a clear upward trend in the acceptance of nuclear energy in Poland.

According to the same CBOS analysis, this upward trend might stem from the perception of the threat of climate change, as a significant majority of Poles (77%) view climate change as a threat, with over half (51%) considering it a significant dangerous phenomena. Simultaneously, 57% believe that the government should do more to prevent climate change, and 67% are willing to take individual actions to protect the environment [38]. The threat's significance is most frequently emphasized by respondents from cities with over 500,000 inhabitants (48%) and higher education, (88%). Political orientation also plays a significant role in perceiving climate change, with 41% of left-leaning respondents and 16% of right-leaning respondents identifying it as one of the most significant civilizational threats.

Recent research by Bodara et al. [19] and Bohdanowicz et al. [20] in Poland challenges the assumption that greater climate change awareness leads to increased nuclear energy acceptance. Their findings suggest a negative correlation between pro-environmental attitudes and nuclear energy acceptance. Interestingly, this negative correlation is smaller in Poland than in Western European countries [20]. Building upon the CBOS analysis, we hypothesize that the relationship between climate change attitudes and energy acceptance is present but is moderated by political ideology. This is due to the observed disparities in energy acceptance and climate change attitudes between conservatives and liberals.

## Aims

We aim to provide a clear understanding of public attitudes towards nuclear energy in Poland. As nuclear energy becomes more important in addressing global energy needs, it is essential to find new methods of studying public attitudes towards it. We believe that detailed acceptance studies are necessary to tailor appropriate strategies for policy implementation and increase overall policy acceptance.

Our research aims to:

- **Aim 1**: Investigate how the complex network of attitudes, societal views and environmental views interconnect to influence current public acceptance of nuclear energy in Poland. However, while focusing exclusively on nuclear energy is informative, it may not provide a complete picture. As highlighted in our literature review and supported by Klein et al. [16], there seems to be a polarization between those who support nuclear energy and those who advocate for renewables. We seek to expand our scope by also incorporating attitudes towards renewable energy into our analysis, aiming to provide a more holistic view.

- **Aim 2**: Establish what drives public support for nuclear energy. Specifically, we measure public acceptance through the lens of investment attitude and perceived advantageous trade-off. We ask respondents whether they think expanding the share of nuclear energy in the energy mix is a good investment and whether the potential benefits of nuclear energy out-weigh associated risks. We use these two questions as proxies for the acceptance of nuclear energy and determine what other beliefs are associated with respondent answers to the two questions.

- **Aim 3**: Assess the different strategies governments and organizations can employ to shape public opinion about nuclear energy and climate issues. More than just reviewing these strategies, we hope to compare them, offering insights for policymakers about what works best when communicating with the public and potential challenges each strategy may face.

By interlinking these facets, we aim to present the most thorough insight into public acceptance of nuclear energy to date. Ultimately, we hope our research aids in crafting well-informed and impactful future policies.

## Materials and methods

### Acceptance models

Studies on public attitudes toward climate issues and energy source acceptance have traditionally relied on presenting statistics describing the proportion of a particular belief within a studied sample. This form of data presentation is common for government reports or surveys that play a significant role in shaping energy policy (e.g.: 54% of respondents declare support for the construction of nuclear power plants in Poland). However, while proportions are easy to interpret, they ignore the complex relationships that exist in the data.

To address this limitation, structural models of acceptance have been used to represent causal relationships between specific attitudes and energy source acceptance [39, 40]. While structural models are useful in representing cause-and-effect relationships, they are limited by the number of attitudes that can be analyzed with them. Moreover, structural models may be inadequate in representing complex patterns of dependencies and often require dimension reduction methods, such as factor analysis, which may result in a significant loss of information and assumptions that are difficult to accept in public attitude research [41].

Our article introduces a novel method for analyzing social attitudes and acceptance, based on causal attitude networks [41]. Our goal is to visualize attitudes in the form of a graph, where nodes represent beliefs related to the climate and energy sources and edges denote relationships or co-occurrences between these beliefs. The network structure reveals which beliefs are most likely to co-occur and which are most central in shaping climate attitudes and public acceptance.

Network analysis is a popular method in sociological [42], psychological [43], and political science [44] research. Previous studies have used network models to analyze political beliefs and candidate acceptance in elections [45], attitudes towards GMO [46] and beliefs related to COVID policies [47]. We are the first to employ exploratory network analysis to investigate climate and energy source attitudes and evaluate various policy strategies.

### Conceptualization of attitudes

It is possible to conceptualize many attitudes concerning climate issues and energy source acceptance. We identified the most important attitudes people can have about climate change and energy sources by carefully reviewing the literature (see literature review) and major institutional surveys (e.g. Eurobarometer). We then operationalized these attitudes into single-

**Table 1. Climate change beliefs and their associated proportions.**

| Construct Name | Supporting Belief | Opposing Belief | Proportion Support |
|---|---|---|---|
| **Climate change beliefs** | | | |
| Urgency | Climate change requires immediate action. | Climate change does not require spending on countermeasures. | 55.92% |
| Conservationism | The preservation of natural environments takes precedence over potential price increases for goods and services. | Human utilization of the natural environment can be unrestricted. | 51.48% |
| Conventional energy | We should aim to completely phase out conventional energy sources such as wood, coal, gas, and oil. | We should continue to invest in conventional energy sources. | 50.59% |
| Progressivism | The state should support a transition to green energy sources. | The state should not support a transition to green energy sources. | 49.70% |
| Centralization | The energy market should be centralized (managed by state-owned companies). | The energy market should be decentralized (managed by private companies). | 43.49% |

**Note**: *Supporting Belief* corresponds to a positive attitude towards a given construct and is represented as +1 in our data and Ising model. *Opposing Belief* represents a negative view of the construct and is represented as -1. The labeling is arbitrary. *Proportion Support* indicates the proportion of total responses that declared a supporting belief (i.e., a positive attitude coded as +1).

answer items that represent a person's stance on a given issue. According to the causal attitude network literature [41], attitudes exist on a spectrum, with supporting (pro) beliefs on one side and opposing (against) beliefs on the other. We aimed to reflect this spectrum in the framing of our items, where respondents were asked to choose their attitude from opposite beliefs. For example, a person could choose between the belief that nuclear energy is a good investment or the belief that it is a bad investment, reflecting their supporting or opposing attitude towards nuclear energy acceptance.

In our data and subsequent analysis, a supporting belief corresponds to a positive attitude towards a given problem and is represented as +1, whereas an opposing belief represents a negative attitude and is represented as -1. The attitudes were grouped into three categories: Climate Change Beliefs, which includes attitudes towards the urgency of climate change; Nuclear Energy Beliefs, and Renewable Energy Beliefs, each category containing the same set of attitudes such as the investment attitude which evaluates whether investing in nuclear or renewable energy is perceived as good or bad. Additionally, political orientation was measured by self-reported conservative and liberal beliefs.

Below, Tables 1 and 2 present a list of 13 attitudes that we identified as relevant when considering climate issues and the energy source acceptance.

## Network model

In our study, we operationalized beliefs as binary variables, representing opposing attitudes, which led us to implement the Ising model as a causal attitude network. The Ising model is a model for pairwise interactions between binary variables and was originally developed for ferromagnetism research [48]. In physics it is used to model the alignment between positive (1) and negative (-1) atom states, but atoms can be replaced by any variable of substantial meaning. The model assumes that the alignment of the current configuration influences the likelihood of its occurrence. Alignment occurs when two neighboring variables reach the same state (either 1 or -1), while divergence occurs when neighbors have different states (i.e., -1, 1 or 1, -1). We coded beliefs supporting nuclear and renewable energy and pro-environmental attitudes as +1, and those opposing as -1. We also incorporated political beliefs, assigning +1 to left-wing and -1 to right-wing views.

**Table 2. Energy source beliefs and their associated proportions.**

| Construct Name | Supporting Belief | Opposing Belief | Proportion Supporting | |
|---|---|---|---|---|
| | | | Renewables | Nuclear |
| **Energy source beliefs** | | | | |
| Risk (risk-benefit tradeoff) | The potential benefits associated with [energy source] outweigh its risks. | The potential risks associated with [energy source] outweigh its benefits. | 83.43% | 49.41% |
| Efficiency | [Energy source] is efficient enough to supply most of the national energy demand. | [Energy source] is not efficient enough to supply most of the national energy demand. | 51.48% | 73.37% |
| Waste safety | In the case of [energy source], the utilization of waste [examples of waste] is safe for the environment. | In the case of [energy source], the utilization of waste [examples of waste] is dangerous for the environment. | 47.33% | 42.01% |
| Global warming | [Energy source] plays a crucial role in mitigating global warming issues. | [Energy source] exacerbates global warming issues. | 80.47% | 67.16% |
| Safety | The use of [energy source] is not risky. | The use of [energy source] is risky. | 66.86% | 81.66% |
| Independence | The adoption of [energy source] will increase a country's energy independence. | The adoption of [energy source] will decrease a country's energy independence. | 64.79% | 74.56% |
| Prices | [Energy source] provides lower energy prices. | [Energy source] leads to higher energy prices. | 49.70% | 63.90% |
| Investment | Expanding the share of [energy source] in the national energy mix is a good investment. | Expanding the share of [energy source] in the national energy mix is a bad investment. | 80.77% | 60.95% |

**Note**: The *Proportion Supporting* columns reflect the division of attitudes between renewable energy and nuclear energy. Each proportion is calculated as the percentage of total responses that support the belief (i.e., a positive attitude towards the specified energy source) for each construct separately.

The interpretation of alignment and divergence coefficients depends on how the beliefs were coded. For example, a strong alignment coefficient between beliefs represented as "Nuclear Investment" and "Nuclear Safety" means that, on average, individuals who believe that nuclear energy is a good (bad) investment also consider it a safe (dangerous) energy source. Conversely, a strong divergence coefficient would suggest that beliefs tend to co-occur as opposites (i.e. bad (good) investment and safe (dangerous) source). To further illustrate, if "Political orientation" scores +1 (indicative of liberal beliefs) and "Nuclear Investment" scores -1, this divergence suggests that, on average, liberals might view nuclear energy as a bad investment, whereas conservatives might see it as beneficial. A network coefficient of 0 between two nodes implies no relationship between the beliefs, indicating their independence.

Additional details on the Ising model, including the R code for calculating the model and accessing its coefficients, can be found in the S1 File.

## Survey methodology and sampling

In December 2022, we conducted a survey on a nationally representative sample of 338 Polish individuals, selecting the sample through a randomized process and stratifying by age to match the age structure of the Polish population, as detailed in Table 3. The recruitment period for our study began on December 10, 2022, and concluded on December 20, 2022. Each participant was briefed at the start of the interview about the study's nature, objectives, and data processing techniques, with an assurance of data anonymity. We employed the Computer-Assisted Telephone Interviewing (CATI) technique combined with Random Digit Dialing (RDD) for data collection. Given the nature of CATI, informed consent was obtained verbally; participants' continued engagement after the briefing was taken as their consent to participate. This method aligns with standard ethical practices for telephone-based surveys. Our survey, comprising 22 items in total, included the first six questions to gauge general societal attitudes

**Table 3. Age structure of respondents.**

| Age | Percentage of total population | Number of respondents | Proportion |
|---|---|---|---|
| >64 | 22% | 91 | 27% |
| 60–64 | 7% | 27 | 8% |
| 55–59 | 6% | 24 | 7% |
| 50–54 | 6% | 24 | 7% |
| 45–49 | 7% | 30 | 9% |
| 40–44 | 8% | 34 | 10% |
| 35–39 | 8% | 34 | 10% |
| 30–34 | 7% | 30 | 9% |
| 25–29 | 6% | 24 | 7% |
| 20–24 | 5% | 20 | 6% |
| **Total** | **82%** | **338** | **100%** |

Note: The proportions in the population do not sum to 100% because individuals below 20 years old were excluded from our sample for legal reasons.

and the subsequent 16 questions in two subsections to assess perceptions on nuclear energy and renewable energy using identical questions. We did not include minors in our study.

# Results

## High level results

This paper quantified acceptance of energy sources with respect to both nuclear and renewable energy [Fig 1]. We assessed acceptance using the measures of 'risk-benefit tradeoff' (Risk) and 'continued investment' (Investment). Interestingly, these two measures were found to be uncorrelated with each other. This suggests that decisions concerning 'investment into energy sources' and perceived 'benefits/risks' are not strictly linked [see Fig 1].

Political ideology was found to be the most central determinant in predicting respondent stances on acceptance of energy sources and many other beliefs. The strongest links manifested between political ideology and climate urgency [Fig 1, Top]. Specifically, Liberal views were highly aligned with: perceived urgency (Urgency), increased intent to phase out traditional energy sources (Conventional), a prioritization of environmental protection (Conservationism), increased state control over the energy market (Centralization), and the need for state support of green energy energy (Progressivism).

The beliefs surrounding renewable energy were largely aligned, meaning that respondents who had positive views regarding renewable safety (Safety) also were likely to believe: that renewable waste was not a significant challenge (Waste safety), that renewables were efficient enough to provide most of a country's energy supply (Efficiency), and that renewables would decrease energy prices (Prices) [Fig 1, Right]. These beliefs in turn were also predictably aligned with the previously-mentioned climate-related views such as increased urgency towards climate change (Urgency), increased intent to phase out conventional energy (Conventional), and higher priority of environmental conservation (Conservationism).

In contrast, acceptance of nuclear energy (for both investment and risk tradeoffs) are largely unaligned with other views other than political ideology. The perceived safety of nuclear energy (Safety), the perceived lower prices from the use of nuclear energy (Prices), and efficiency of nuclear energy (Efficiency) are all uncorrelated with both investment and perceived risk tradeoffs (Risk) of nuclear energy [Fig 1, Left].

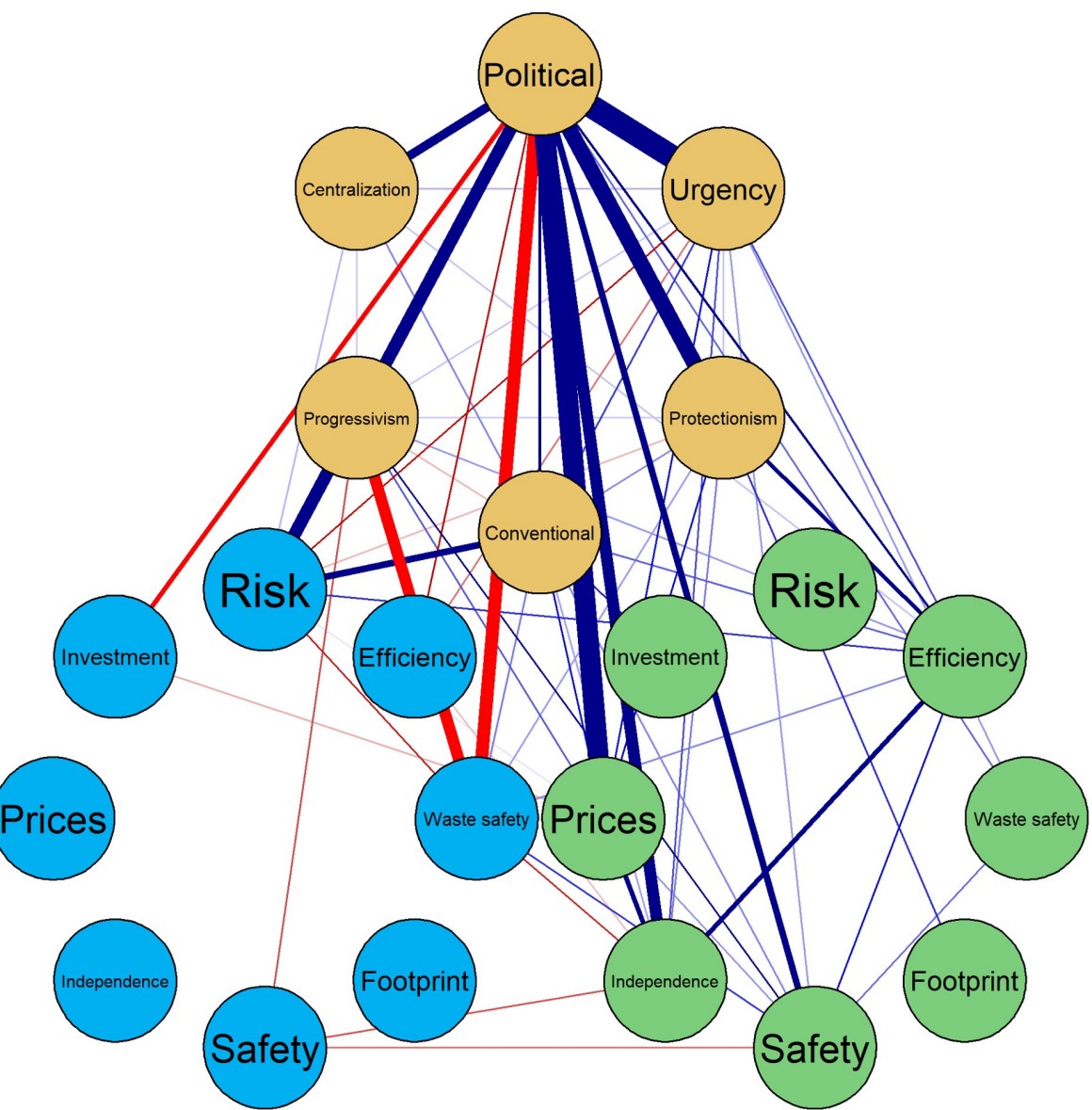

**Fig 1. Network model illustrating attitudes towards climate and energy sources.** Nodes represent beliefs: Climate change beliefs (yellow), nuclear energy beliefs (blue), and renewable energy beliefs (green). Blue edges indicate alignment between beliefs, meaning individuals who hold one belief tend to also hold the other. Conversely, red edges represent divergence, meaning individuals with one belief are likely to have the opposite belief for the other node. The thickness of an edge corresponds to the strength of the alignment or divergence between the connected beliefs. Nodes without an edge between them are independent beliefs. The description of the nodes can be found in Tables 1 and 2.

The perceived tradeoff for nuclear energy is determined through several seemingly conflicting beliefs. For example, while nuclear energy's risk-benefit tradeoff is aligned to the need for the phasing out of conventional energy and increased government control over energy, the tradeoff is divergent with increased priority of environmental conservation and climate urgency [Fig 1, Top and Left].

While respondents with liberal views see a positive risk-benefit tradeoff for nuclear energy, conservatives do not. Conversely, conservative respondents want to continue investment into

nuclear energy while liberal respondents do not. This is a striking difference where the tradeoff and continued investment measures are conversely aligned with conservative political views.

## Granular level results: Conservatives vs liberals

This paper has also analyzed the specific views of respondents concerning a variety of energy source characteristics on a granular level. As the most central variable in the model was political beliefs, we wanted to investigate the association of political beliefs with two variables that are important for nuclear energy acceptance: risk-benefit tradeoff and investment attitude.

**Risk-benefit tradeoff.** Compared to Liberals, Conservative respondents do not see a positive risk-benefit tradeoff for nuclear energy (Fig 2). However, they similarly see nuclear energy as safe to operate (Fig 2.3) and believe that nuclear power increases national energy independence (Fig 2.1). The difference is likely explained by Liberal respondents being significantly more concerned about nuclear waste (Fig 2.4) while simultaneously seeing a need to phase out conventional energy (Fig 2.2). Comparatively few Conservatives see a need to phase out conventional energy or are concerned with nuclear waste (Fig 2.4).

The reasons why Conservatives are more likely than Liberals to perceive a negative risk tradeoff for nuclear energy are unclear. Despite seeing nuclear energy as safe to operate and being less concerned about nuclear waste or the need to phase out conventional energy, Conservatives still perceive a negative risk tradeoff (Fig 2.3). Such results suggest that risk-benefit tradeoffs are not primarily driven by safety concerns.

Liberal respondents likely see a positive risk-benefit tradeoff for nuclear power because they perceive the benefits of nuclear energy, particularly its potential to phase out conventional energy sources, as outweighing the concerns regarding nuclear waste safety. This viewpoint

#### Figure 2.1 Energy independence factor

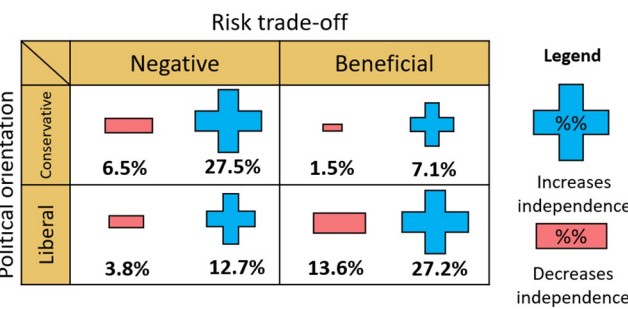

#### Figure 2.2 Conventional energy factor

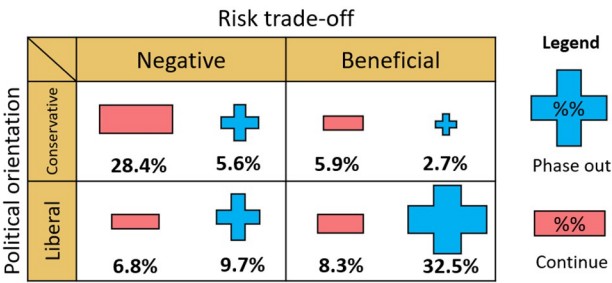

#### Figure 2.3 Nuclear operation safety factor

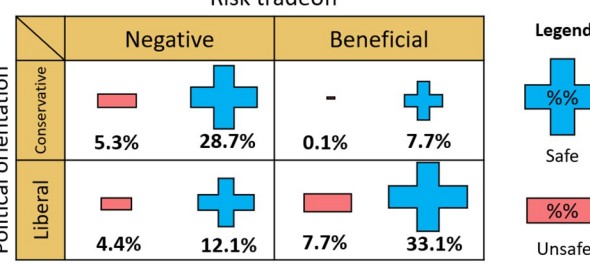

#### Figure 2.4 Nuclear waste safety energy factor

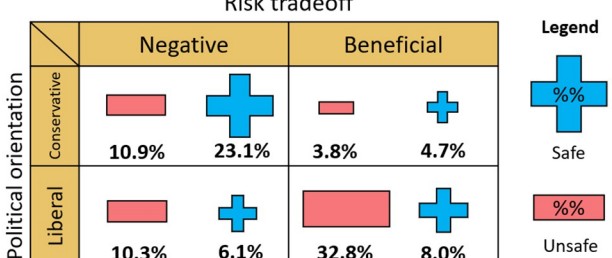

**Fig 2. Aspects determining political risk trade-offs for nuclear energy.** The percentages presented are proportions of answers in each category, with the larger symbols indicating a greater number of responses. The given proportions are rounded for interpretability, and the percentages for each part of the figure sum up to 100%, accounting for any rounding errors. For detailed response rate numbers, refer to the contingency tables in the S1 File.

#### Figure 3.1 Energy efficiency factor

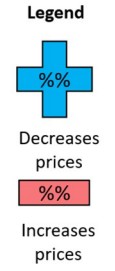

#### Figure 3.2 Conventional energy factor

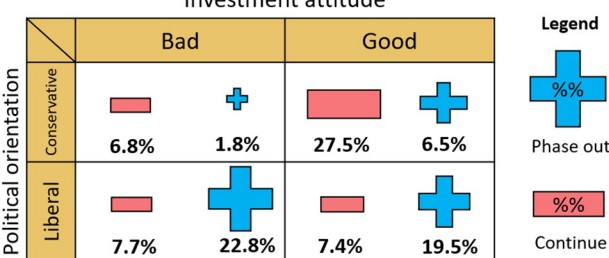

#### Figure 3.3 Energy prices factor

#### Figure 3.4 Nuclear waste safety energy factor

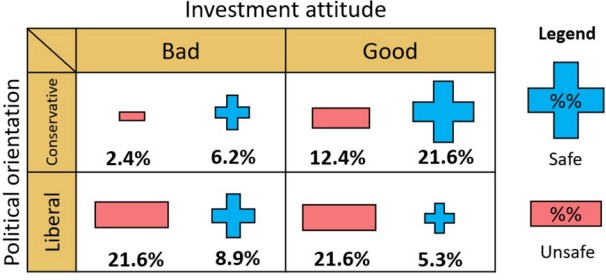

**Fig 3. Aspects determining political investment acceptance of nuclear energy.** For detailed response rate numbers, refer to the contingency tables in the S1 File.

aligns with their broader concerns about global warming and the need to reduce reliance on conventional energy sources (Fig 2.2).

**Investment attitude.** In contrast to Liberals, Conservative respondents significantly support increased investment in nuclear energy (Fig 3). This support is even more surprising given their perception of a negative risk tradeoff. Most Conservatives see nuclear power as efficient (Fig 3.1), believe that nuclear waste is not a significant problem (Fig 3.4), and similarly see nuclear as decreasing prices (Fig 3.3). This overwhelming support for continued nuclear investment (Fig 3) with 80% of Conservatives supporting investment compared to 45% of Liberals, suggests that investment attitudes are likely driven by political motivations rather than a rational interpretation of the risk-benefit tradeoff or economic factors such as prices.

The strong Conservative support for nuclear investment probably stems from the perceived efficiency of nuclear energy, as there is some relationship between political orientation and views on efficiency. However, other potentially important factors, such as prices, have no effect on investment attitudes and are unrelated to political ideology. This implies that political beliefs play a crucial role in shaping investment attitudes, overshadowing rational considerations of risk-benefit tradeoffs and economic factors.

Additional statistical comparisons between Liberal and Conservative responses are provided in the S1 File.

## Discussion

### Discussion of results

This paper's results support past findings, contradict some past conjectures, and engender interesting questions concerning the nature of nuclear acceptance.

**Political ideology.** The high-level findings show that the most central factor in nuclear acceptance, whether measured as a risk-benefit tradeoff or continued investment, is political ideology. Such a result is not unexpected as past studies such as Chung et al. [12] have found political affiliation to be significant for predicting a person's preference for nuclear power. Past research found that voters adopt their political party's preference rather than generating their own views based on their knowledge [10, 13].

In addition, there are several mechanisms identified, such as by Qi et al. [7], which show that trust in authorities is the central determinant in impacting public view. Here, voter trust in their ideological leaders (and therefore leader messages or stances towards nuclear power) is likely responsible for the strong significance of political ideology as a variable. Ideological affiliation and source trust has proved key in other papers as well [22].

Views on nuclear energy can be either generated independently, acquired through an incorporation of an ideology as personal beliefs, or acquired from trusted political leaders. These two latter pathways likely explain why political ideology is such a strong indicator of nuclear acceptance views.

While political ideology was found to be significant in predicting nuclear acceptance, its individual predictions are diametrically opposed for nuclear investment and nuclear tradeoff views. This result challenges the notion that willingness to invest and perceived risk-benefit are aligned. In other words, voters who support continued investment are not necessarily positively predisposed to nuclear power (see a beneficial tradeoff) and those who are positively predisposed do not necessarily view nuclear power as a good investment. Around $\frac{1}{3}$ of respondents for both conservative and liberal ideologies do not have the two answers aligned (e.g. good investment but bad tradeoff). The conservative respondents see nuclear energy as a bad risk-benefit tradeoff but a good investment while liberal respondents see nuclear energy as a beneficial tradeoff but bad investment.

**Climate change.** Climate urgency is itself a central node, being positively associated with many pro-renewable energy attitudes, but also negatively linked with attitudes towards nuclear energy risk tradeoff and efficiency. This negative link is a replication of findings of Bohdanowicz et al. [20]. However, our study offers further insight into the underpinnings of this relationship. We found that as people who want immediate climate actions (Urgent) perceive nuclear energy as inefficient (Efficiency) and not beneficial due to its high cost and long-term investment nature, consequently gravitating towards more immediate solutions, i.e. renewables.

**Conflicting beliefs.** Our results represent another data point in the new wave of research that challenges traditional beliefs. Sjoberg et al. and Temper et al. [23, 24] found that perceived nuclear energy risks do not directly influence respondent attitude. Bickerstaff et al. and Hu at al. [17, 34] found that endorsers of nuclear energy are often knowledgeable about climate change and engaged in environmental issues. Zeng et al. [49] found that respondents who were more knowledgeable about nuclear power were often exposed to negative information and therefore observed higher risks (challenging the notion that more teaching improves acceptance).

This paper presents similarly surprising results. Views around renewable energy are aligned (ex: respondents who see renewables as increasing country independence also tend to see renewables as decreasing prices and having strong efficiency to provide a country all of its electricity). However, these links do not exist amongst beliefs surrounding nuclear energy. Respondent views on whether 'nuclear energy decreases prices', whether it 'increases independence', or whether 'using nuclear energy is safe' are all uncorrelated with each other just as the beliefs 'nuclear energy is a good tradeoff' and 'good investment' are uncorrelated.

## Discussion of strategies for policy implementation

Given the results of our analysis, we attempt to evaluate different strategies that governments or organizations may adopt when aiming at increasing nuclear energy acceptance.

**Risk-risk trade-off strategy.**   Risk-risk trade-off strategy, rooted in utility theory, involves weighing the potential risks of climate change against the risks from nuclear power plants and radioactive wastes [25]. This strategy communicates to the public that the relatively minor risks associated with operating a nuclear power plant are outweighed by the substantial and inevitable risks of climate change, thus constructing nuclear power plants to combat climate change represents the optimal solution. Initially, this strategy was used as an argument for reshaping public acceptance after the Chernobyl disaster [34]. However, our research, along with studies from Korea [12] and China [49], suggests that this strategy may no longer be as effective. While a substantial portion of the Polish population views nuclear energy as more risky than beneficial (50%), this trade-off is not driven by the negative perception of the safety of nuclear power plants, as a majority of respondents consider them safe (82%). Interestingly, a large number of those who perceive nuclear energy as having a negative trade-off also regard it as a good investment.

The assumption that operational risk (safety) is central for acceptance of nuclear energy is unsupported. While individuals living near nuclear power plants may still harbor negative attitudes towards operation safety, as evidenced by the "not in my backyard" effect, this group is relatively small and does not represent the overall societal acceptance. Furthermore, implementing a risk-risk trade-off strategy may inadvertently reduce acceptance by drawing attention to the risk-factor associated with nuclear plant operation.

**Interest-focused and technology-focused strategies.**   Hu et al. [17] proposed two strategies for improving public acceptance of nuclear energy: the interest-focused strategy and the technology-focused strategy. The former emphasizes the benefits of nuclear power, while the latter seeks to educate the public about the technical and scientific aspects of nuclear energy.

Governments often use interest-focused campaigns, emphasizing nuclear technology's role in energy independence, to boost public acceptance. Poland's conservative government adopted this strategy for its energy policy. Yet, our findings indicate that political views, more than beliefs about safety and independence, drive nuclear energy acceptance. This aligns with global trends showing a correlation between conservative views and acceptance of nuclear energy, and liberal views and rejection of nuclear energy [11–13]. Yeo et al. [11] suggested that people form opinions about climate and attitudes towards energy solutions based on simple heuristics, rather than calculations of potential risks and rewards.

The technology-based strategy, which focuses on presenting independent scientific knowledge, may be a better approach. This strategy is reminiscent of the deficit model [50], which suggests that negative public attitudes arise from a deficit of knowledge and education. However, this approach has limitations; fostering acceptance through education is time-consuming and may not appeal to all, particularly with subjects as complex and politicized as nuclear technology. Moreover, if education campaigns are associated with government and political views, they again risk being perceived as propaganda rather than reliable sources of knowledge, leading back to the same problem faced by the interest-based strategy [51].

**Framing strategy.**   Framing is a strategy that combines the strength of interest-based strategy while circumventing the pitfalls of the association with political ideology. Framing refers to "tailoring messages to the existing attitudes, values, and perceptions of different audiences" [52].

The advantage of this strategy is that we first identify important attitudes and values for a group that has low acceptance of nuclear energy, and then present nuclear energy through

those attitudes and values. For example, presenting nuclear energy as bolstering national independence and economic strength may attract conservatives, while emphasizing its environment-friendly nature might appeal to liberals. Interestingly, many liberals favor nuclear power over conventional sources like coal and contrasting nuclear with these sources improves its environment-friendly image, aligning nuclear benefits with specific political values: national strength for conservatives and environmentalism for liberals.

The European Commission has utilized the green energy frame by labeling nuclear energy as "green energy" in their new EU taxonomy regulation. This framing strategy positions nuclear energy alongside renewables in public discourse, fostering a perception of belonging to the same category. An extension of this strategy could involve presenting nuclear energy alongside renewables in public campaigns promoting climate change solutions and media.

Previous research has shown that reframing nuclear energy can be somewhat effective [25, 34, 53]. However, the success of a frame largely depends on its alignment with the audience's existing attitudes. If a frame uses inappropriate references (like a nationalistic frame for a liberal audience) or utilizes beliefs already widespread among the population (such as the economic benefits), the frame's impact will be minimal or non-existent [35]. Framing strategies for groups that already hold largely positive views towards a specific energy solution, such as conservatives towards nuclear energy and liberals towards renewables, may not be effective. Implementing a campaign or policy with an inappropriate frame or for an already high acceptance group could lead to wasted resources or in the worst scenario decreased public acceptance. This highlights the importance of comprehensive studies, like ours, that examine nationwide acceptance. These studies help researchers and policymakers spot discrepancies in public attitudes and apply the most effective frames.

## Conclusion

Our research aimed to investigate the network of public attitudes towards nuclear energy in Poland and compared them to those for renewable energy. We pioneered a unique network approach to assess public acceptance, enabling us to gauge the complex relationship between energy source acceptance, political ideology, perceived risks and benefits, and broader societal and environmental attitudes. Among our most salient findings is the central role of political ideology in determining public attitudes. We found that liberals align more with the urgency of climate change, phasing out traditional energy sources, prioritizing environmental protection, increasing state control over the energy market, and supporting renewable energy. Conservatives, on the other hand, generally support continued investment into nuclear energy.

Nevertheless, some inconsistencies in attitudes were evident. For instance, while conservatives largely view nuclear power as a viable investment and acknowledge its operational safety and role in energy independence, they also see its risk-benefit tradeoff in a negative light. This inconsistency hints at an underlying cognitive or informational dissonance that needs further examination. Similarly, most Liberal respondents do not support continued investment into nuclear energy. However, they see nuclear energy as efficient and believe it will decrease energy prices.

These findings contribute a novel perspective to the existing research on energy acceptance. Our approach, distinguished by its detailed analysis, is particularly relevant for social change strategy planning. As discussed, many popular government policies centered on risk-risk tradeoff and interest-based strategies fall short. In contrast, framing strategies, which can integrate insights from acceptance studies, seem more promising. They can pinpoint and engage beliefs and attitudes likely to enhance acceptance. In the current politically polarized context,

this might mean utilizing a "green" framing of nuclear energy for liberals and an emphasis on energy independence for conservatives.

## Supporting information

**S1 File. The supplementary materials are accessible via the following link: https://osf.io/neagr/.** The supplementary materials include: Dataset: This contains the raw data used in the analysis. Supplementary explanations: This document provides a breakdown of the code and additional explanations related to the model used in the analysis. Supplementary tables: This document contains supplementary tables with additional data and information related to Figs 2 and 3 in the main text. R script: This is the R script containing the code used for data analysis and generating the results and figures.
(ZIP)

## Author Contributions

**Conceptualization:** Pawel Robert Smolinski, Joseph Januszewicz, Barbara Pawlowska, Jacek Winiarski.

**Data curation:** Pawel Robert Smolinski.

**Formal analysis:** Pawel Robert Smolinski, Joseph Januszewicz.

**Funding acquisition:** Jacek Winiarski.

**Investigation:** Pawel Robert Smolinski.

**Methodology:** Pawel Robert Smolinski.

**Project administration:** Pawel Robert Smolinski, Joseph Januszewicz.

**Resources:** Pawel Robert Smolinski.

**Software:** Pawel Robert Smolinski.

**Supervision:** Pawel Robert Smolinski, Joseph Januszewicz.

**Validation:** Pawel Robert Smolinski, Joseph Januszewicz.

**Visualization:** Pawel Robert Smolinski.

**Writing – original draft:** Pawel Robert Smolinski, Joseph Januszewicz, Barbara Pawlowska, Jacek Winiarski.

**Writing – review & editing:** Pawel Robert Smolinski, Joseph Januszewicz, Barbara Pawlowska, Jacek Winiarski.

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
