## [Decision Letter · Decision Letter 0]

22 Mar 2024

PONE-D-23-38813Nuclear Energy Acceptance in Poland: From Societal Attitudes to Effective Policy Strategies - Network Modeling ApproachPLOS ONE

Dear Dr. Winiarski,

Thank you for submitting your manuscript to PLOS ONE. After careful consideration, we feel that it has merit but does not fully meet PLOS ONE’s publication criteria as it currently stands. Therefore, we invite you to submit a revised version of the manuscript that addresses the points raised during the review process.

We look forward to receiving your revised manuscript.

Kind regards,

Mohammad Alrwashdeh

Academic Editor

PLOS ONE

Journal Requirements:

3. Did you know that depositing data in a repository is associated with up to a 25% citation advantage (https://doi.org/10.1371/journal.pone.0230416)? If you’ve not already done so, consider depositing your raw data in a repository to ensure your work is read, appreciated and cited by the largest possible audience. You’ll also earn an Accessible Data icon on your published paper if you deposit your data in any participating repository (https://plos.org/open-science/open-data/#accessible-data).

4. You indicated that ethical approval was not necessary for your study. We understand that the framework for ethical oversight requirements for studies of this type may differ depending on the setting and we would appreciate some further clarification regarding your research. Could you please provide further details on why your study is exempt from the need for approval and confirmation from your institutional review board or research ethics committee (e.g., in the form of a letter or email correspondence) that ethics review was not necessary for this study? Please include a copy of the correspondence as an ""Other"" file.

This research was funded by the University of Gdansk, Faculty of Economics. Internal financing

7. We are unable to open your Supporting Information file figure1.svg. Please kindly revise as necessary and re-upload.

Reviewers' comments:

Reviewer's Responses to Questions

**Comments to the Author**

1. Is the manuscript technically sound, and do the data support the conclusions?

Reviewer #1: Yes

Reviewer #2: Yes

Reviewer #3: Yes

Reviewer #4: Yes

2. Has the statistical analysis been performed appropriately and rigorously? 

Reviewer #1: Yes

Reviewer #2: Yes

Reviewer #3: Yes

Reviewer #4: Yes

3. Have the authors made all data underlying the findings in their manuscript fully available?

Reviewer #1: Yes

Reviewer #2: Yes

Reviewer #3: Yes

Reviewer #4: Yes

4. Is the manuscript presented in an intelligible fashion and written in standard English?

Reviewer #1: Yes

Reviewer #2: Yes

Reviewer #3: Yes

Reviewer #4: Yes

5. Review Comments to the Author

**Reviewer #1**: I find the study interesting and would like to recommend few minor changes in the manuscript to make it easier for the community to follow.

1. I would recommend that, for a better understanding, include a simple example of causal attitude networks. That would make it easier for the readers to follow.

2. Page 17, rather than the term "creation" for Nuclear power plant, My suggestion is to use "construction".

**Reviewer #2:** This paper investigates public attitudes and acceptance of nuclear energy in Poland using a novel network modeling approach. The authors surveyed 338 Polish participants to understand the complex relationships between political ideology, environmental attitudes, risk perceptions, safety concerns, and economic factors in shaping acceptance of nuclear energy compared to renewable energy. The key findings are:

• Political ideology is the most central factor influencing acceptance of nuclear energy. Liberals align more with climate change urgency and support for renewables, while conservatives generally support investment in nuclear energy.

• Environmental attitudes, risk perceptions, safety concerns, and economic variables also play substantial roles in nuclear energy acceptance, but in complex and sometimes contradictory ways.

• Beliefs about renewable energy are more aligned and correlated compared to beliefs about nuclear energy, which are more fragmented.

• Strategies to increase nuclear acceptance, like risk-risk tradeoff and technology-focused approaches, may have limitations. The authors advocate for a framing strategy that tailors messages to distinct societal values.

Overall the paper is appropriate for publication in PLOS ONE. It covers an important, policy-relevant topic for decarbonizing the power sector in a country dominated by emissions-intensive coal-based generation. The overall methodology is clear and appropriate. Evaluation based on PLOS ONE criteria:

• The study presents the results of primary scientific research. It uses original survey data and a novel network analysis methodology.

• To my knowledge, the specific results reported have not been published elsewhere. The authors build upon and reference related prior research.

• The survey methodology and network analysis are robust. Sufficient details are provided on the sample, survey questions, and analytical approach.

• The conclusions are presented appropriately and supported by the data and analysis. The authors discuss the findings in the context of energy policy implications.

• The article is well-written in standard English and presented in an accessible form, with clear structure and explanations of concepts and methods.

• The research appears to meet applicable standards for the ethics of experimentation and research integrity. The authors provide an ethics statement on the survey methodology. However, I have a comment to be addressed (provided later in the review).

• The article adheres to appropriate reporting guidelines and standards for data availability.

• The authors state the survey data and R code are accessible via a provided link. However, permissions are required to access the data.

In overall consideration of the work, I do find this paper interesting, and potentially worthy of publication. This said, some significant revisions are requested. These revisions are as follows:

• I am a bit concerned to see that institutional ethical review was not conducted for the survey questions and approach. I would like the authors to elaborate the university standard policies on this.

• Figure and Table captions – I cannot find figure and table captions in the review material. These must be provided.

• Table 1 – the information content is confusing. Specifically the information presented on “proportion supporting” need to be elaborated in the text or in explanatory material associated with the table.

• Figures 2 and 3 – I appreciate the author’s effort to show 3 dimensions of information in 2D figures. However, the information took me a while to understand, particularly with figure captions available. The most important issues to address is for the “+” and “-“ symbols. Although size is meant to represent response rate, I need to also see the specific response rate numbers. This should at least be available in supplementary material.

• Statistical analyses – perhaps the most important issue to address, and the cause for requested major revision, is statistical analysis/confirmation of differences between survey response rates. Figures 2 and 3 provide a number of response rates but without any information regarding true statistical differences. Appropriate statistical tests are needed in supplementary material for all cases when comparisons are being made concerning response rates. As an example, the statement on page 22 “most Liberals do see nuclear energy as efficient enough to supply most of a nation’s energy [3.1] and believe that nuclear energy will decrease energy prices if introduced [3.3].” cannot be verified given the information provided currently. Similarly, on page 30 I see the statement “Similarly, most Liberal respondents do not support continued investment into nuclear energy. However, they see nuclear energy as efficient and believe it will decrease energy prices.” However, in Figure 3 I question whether “most Liberal respondents do not support continued investment into nuclear energy” is a valid statement statistically. I need to see the data.

**Reviewer #3:** The authors used a novel network approach to investigate Polish participants on their social attitudes towards various energy sources. The results suggest that political ideology is a central factor influencing energy attitudes, and that factors such as environmental attitudes, risk perceptions, security concerns, and economic variables are also important and have some promising applications.

(1) Paragraph 30 of the conclusion mentions that ’Conservatives, on the other hand, generally support continued investment into nuclear energy.’ The conclusions are overstated. Please explain the meaning of ‘Conservatives’.

(2) In section 2.4 , ’ found that many anti-nuclear protesters worldwide primarily highlight issues related to waste management and environmental pollution’ and ’the emphasis on risk and safety might not be as influential in determining public acceptance as previously believed.’ Why the emphasis is ‘not’ be influential?

(3) Please explain the meaning of ‘the lens of investment attitude and perceived advantageous trade-off.’

(4) Please consider to add some reference about the design of new generation nuclear reactor. Such as:

[1] Zou, J. , Liu, S. , Jin, C. , Chen, Y. , Cai, Y. , & Wang, L. . (2023). Optimization method of burnable poison based on genetic algorithm and artificial neural network. Annals of nuclear energy(Nov.), 192.

[2] Wu, Y. , Liu, S. , Li, M. , Xiao, P. , & Chen, Y. . (2020). Monte carlo simulation of dispersed coated particles in accident tolerant fuel for innovative nuclear reactors. International Journal of Energy Research.

[3] Wang, J., Liu, S., Li, M., **ao, P., Wang, Z., Wang, L., ... & Chen, Y. (2021). Multiobjective genetic algorithm strategies for burnable poison design of pressurized water reactor. International Journal of Energy Research, 45(8), 11930-11942.

**Reviewer #4:** Would you find here below my comments:

page 3: "typical lifespan of a nuclear power plant, which could exceed 80 years." is more appropriate since no plant have yet obtained a license to operate more than 80 years (even if this is considered).

page 4: "we consider factors like political ideology, safety concerns" as well as "environmental and economic variables."

page 8:"Recent surveys conducted by CBOS" indicate what CBOS is when first used in the text.

page 19: "with respect to both" and "nuclear and renewable energy" (and to be deleted)

page 19 and Figure 1: Figure caption not provided on the PLOS ONE page. Is it included in the article. Figure caption and brief description of the figure, in the text, is needed.

page 20: "Political ideology was found to be the most central determinant in predicting respondent stances on acceptance of energy sources and many other beliefs." is the main conclusion of paper. However, according to the data provided, 144 respondents defined themselves as conservative, out of 338 respondents. This corresponds to 42.6 % of the respondents. This value should be mentioned in the paper and commented. Is this proportion in agreement with the proportion of conservatives in Poland at the time of the survey? In other terms is the sample representative? Potentially, the perception of being conservative or liberal may not be related to politics (and political leaders) but to a personal definition of the respondents. What would be the impact on the conclusions of the paper?

page 21: Would you find here below my comments:

page 3: "typical lifespan of a nuclear power plant, which could exceed 80 years." is more appropriate since no plant have yet obtained a license to operate more than 80 years (even if this is considered).

page 4: "we consider factors like political ideology, safety concerns" as well as "environmental and economic variables."

page 8:"Recent surveys conducted by CBOS" indicate what CBOS is when first used in the text.

page 19: "with respect to both" and "nuclear and renewable energy" (and to be deleted)

page 19 and Figure 1: Figure caption not provided on the PLOS ONE page. Is it included in the article. Figure caption and brief description of the figure, in the text, is needed.

page 20: "Political ideology was found to be the most central determinant in predicting respondent stances on acceptance of energy sources and many other beliefs." is the main conclusion of paper. However, according to the data provided, 144 respondents defined themselves as conservative, out of 338 respondents. This corresponds to 42.6 % of the respondents. This value should be mentioned in the paper and commented. Is this proportion in agreement with the proportion of conservatives in Poland at the time of the survey? In other terms is the sample representative? Potentially, the perception of being conservative or liberal may not be related to politics (and political leaders) but to a personal definition of the respondents. What would be the impact on the conclusions of the paper?

page 21: The analysis in section 7.2 contains a number of redundancies. Indeed, the paragraphs "conservatives" and "liberals" include comparisons which are also addressed in the paragraph "political comparison". This section should be revised to suppress these redundancies.

page 24: "These two" latter "pathways likely explain why political ideology is such a strong indicator of nuclear acceptance views." (add latter)

6. PLOS authors have the option to publish the peer review history of their article (what does this mean?). If published, this will include your full peer review and any attached files.

Reviewer #1: No

Reviewer #2: No

Reviewer #3: **Yes:**

Reviewer #4: No

---

## [Author Response · Author response to Decision Letter 0]

22 May 2024

We have attached all responses to the reviewers in the Word file titled "12_05_2024 PLOS rebuttal letter.docx" along with other submission files.

---

## [Editor Report · Decision Letter 1]

24 May 2024

Nuclear Energy Acceptance in Poland: From Societal Attitudes to Effective Policy Strategies - Network Modeling Approach

PONE-D-23-38813R1

Dear Dr. Winiarski,

We’re pleased to inform you that your manuscript has been judged scientifically suitable for publication and will be formally accepted for publication once it meets all outstanding technical requirements.

Kind regards,

Mohammad Alrwashdeh

Academic Editor

PLOS ONE

Additional Editor Comments (optional):

Accepted.
---

## [Editor Report · Acceptance letter]

19 Jun 2024

PONE-D-23-38813R1 

PLOS ONE

Dear Dr. Winiarski, 

I'm pleased to inform you that your manuscript has been deemed suitable for publication in PLOS ONE. Congratulations! Your manuscript is now being handed over to our production team.

Kind regards, 

on behalf of

Dr. Mohammad Alrwashdeh 

Academic Editor

PLOS ONE